# A Scalable Approach to Probabilistic Neuro-Symbolic Robustness Verification

**Vasileios Manginas**                                    VMANGINAS@IIT.DEMOKRITOS.GR
*Institute of Informatics and Telecommunications, NCSR "Demokritos", Greece*

**Nikolaos Manginas**                                     NMANGINAS@IIT.DEMOKRITOS.GR
*Department of Computer Science and Leuven.AI, KU Leuven, Belgium*
*Institute of Informatics and Telecommunications, NCSR "Demokritos", Greece*

**Edward Stevinson**                                     E.STEVINSON22@IMPERIAL.AC.UK
**Sherwin Varghese**                                   SHERWIN.VARGHESE@IMPERIAL.AC.UK
*Department of Computing, Imperial College London, UK*

**Nikos Katzouris**                                           NKATZ@IIT.DEMOKRITOS.GR
**Georgios Paliouras**                                     PALIOURG@IIT.DEMOKRITOS.GR
*Institute of Informatics and Telecommunications, NCSR "Demokritos", Greece*

**Alessio Lomuscio**                                       A.LOMUSCIO@IMPERIAL.AC.UK
*Department of Computing, Imperial College London, UK*

**Editors:** Leilani H. Gilpin, Eleonora Giunchiglia, Pascal Hitzler, and Emile van Krieken

## Abstract

Neuro-Symbolic Artificial Intelligence (NeSy AI) has emerged as a promising direction for integrating neural learning with symbolic reasoning. Typically, in the probabilistic variant of such systems, a neural network first extracts a set of symbols from sub-symbolic input, which are then used by a symbolic component to reason in a probabilistic manner towards answering a query. In this work, we address the problem of formally verifying the robustness of such NeSy probabilistic reasoning systems, therefore paving the way for their safe deployment in critical domains. We analyze the complexity of solving this problem exactly, and show that a decision version of the core computation is $\text{NP}^{\text{PP}}$-complete. In the face of this result, we propose the first approach for approximate, relaxation-based verification of probabilistic NeSy systems. We demonstrate experimentally on a standard NeSy benchmark that the proposed method scales exponentially better than solver-based solutions and apply our technique to a real-world autonomous driving domain, where we verify a safety property under large input dimensionalities.

## 1. Introduction

Neuro-Symbolic Artificial Intelligence (NeSy AI) (Hitzler and Sarker, 2022; Marra et al., 2024) aims to combine the strengths of neural-based learning with those of symbolic reasoning. Such techniques have gained popularity, as they have been shown to improve the generalization capacity and interpretability of neural networks (NNs) by seamlessly combining deep learning with domain knowledge. We focus on probabilistic NeSy approaches that compositionally combine perception with reasoning: first, a NN extracts symbols from sub-symbolic input, which are then processed by a symbolic reasoning component. They rely on formal probabilistic semantics to handle uncertainty in a principled fashion (Marra et al.,

2024), and are for this reason adopted by several state-of-the-art NeSy systems (Manhaeve et al., 2018; Winters et al., 2022; De Smet et al., 2024).

In order to deploy such systems in mission-critical applications, it is often necessary to have formal guarantees of their reliable performance. In this work, we address the challenge of verifying the robustness of probabilistic NeSy systems, i.e., verifying the property that input perturbations do not affect the reasoning output. Techniques for NN verification are valuable to that end, since they can derive robustness guarantees for purely neural systems. In particular, relaxation-based techniques (Ehlers, 2017a; Xu et al., 2020) can scalably compute bounds for the NN outputs, with respect to input perturbations, which can then be used to assess robustness. Our work is focused on extending such techniques to the NeSy case by propagating these bounds through the probabilistic reasoning layer of a NeSy architecture. As such, we can then provide robustness guarantees for the entire system.

Our contributions are as follows: (a) we study the complexity of solving the probabilistic NeSy verification task exactly and show that a decision version of the core computation involved is $\text{NP}^{\text{PP}}$-complete; (b) In the face of this result, we propose an approximate solution, extending relaxation-based NN verification techniques to the NeSy setting. We show how to compile the entire NeSy system into a single computational graph, which encapsulates both the neural and the symbolic components and is amenable to verification by off-the-shelf, state-of-the-art formal NN verifiers; (c) We validate our theoretical results on a standard NeSy benchmark domain, by empirically demonstrating that our proposed approach scales exponentially better than exact, solver-based solutions. Moreover, we show that our method is applicable to real-world problems involving high-dimensional input and challenging network sizes. We do so by applying our technique to an autonomous driving dataset, where we verify the robustness of a safety property, on top of a neural system consisting of an object detection network and an action selector network. The code is available online[1].

## 2. Background

### 2.1. Probabilistic NeSy Systems

A common aim of probabilistic NeSy AI is to combine perception with probabilistic logical reasoning. We provide a brief overview of the operation of such a system based on Marconato et al. (2024). Given input $\boldsymbol{x} \in \mathbb{R}^n$, the system utilizes a NN, as well as symbolic knowledge K, to infer an output $\boldsymbol{y} \in \{0,1\}^m$. In particular, the system computes $p_\theta(\boldsymbol{y} \mid \boldsymbol{x}; \text{K})$, where $\theta$ refers to the trainable parameters of the NN. This is achieved in a two-step process. First, the system extracts a set of $k$ *latent concepts* $\boldsymbol{c} \in \{0,1\}^k$, through the use of a parameterized neural model $p_\theta(\boldsymbol{c} \mid \boldsymbol{x})$. These latent concept predictions are then used as input to a reasoning layer, in conjunction with knowledge K, to infer $p(\boldsymbol{y} \mid \boldsymbol{c}; \text{K})$.

The setting is straightforward to extend to multiple NNs. In that case, the $i^{\text{th}}$ network from a set E would predict $p_\theta^i(\boldsymbol{c^i} \mid \boldsymbol{x})$, with $\bigcup_{i \in \text{E}} \boldsymbol{c^i} = \boldsymbol{c}$. Consider the running example of Figure 1, where two NNs accept the same image as input and output two disjoint sets of latent concepts. These are then combined to form the input to the reasoning layer in order to output the target $\boldsymbol{y}$.

---

1. https://github.com/EVENFLOW-project-EU/nesy-veri

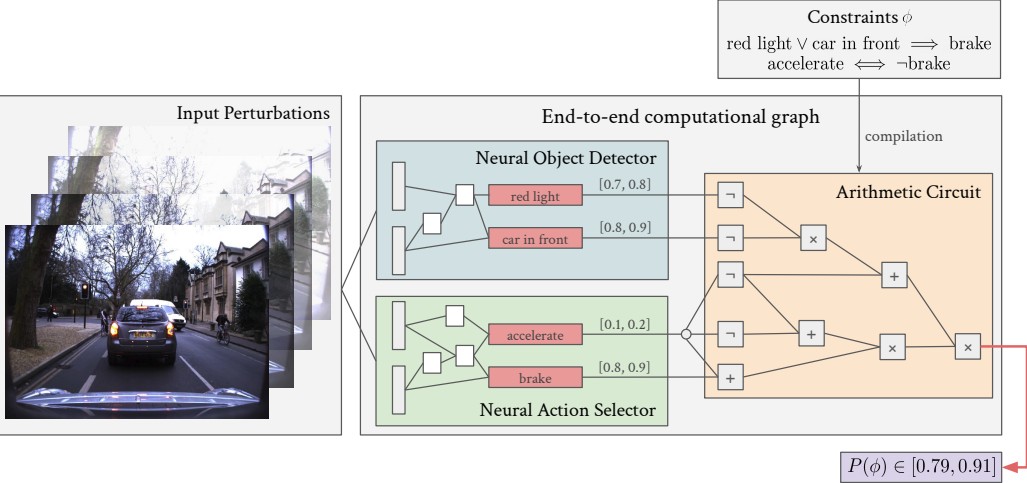

Figure 1: A motivating example for probabilistic NeSy verification. In this autonomous driving example we want to verify two logical constraints $\phi$ on top of two neural networks accepting the same dashcam image as input. The symbolic constraints are compiled into a tractable representation containing only addition, subtraction, and multiplication. During inference, this is used to reason over the NN outputs and calculate the probability that the constraints are satisfied. For verification, we exploit this structure to scalably compute how perturbations in the input affect the probabilistic output of the whole (NNs + reasoning) NeSy system.

### 2.2. Knowledge Compilation

Probabilistic reasoning in NeSy systems is often performed via reduction to Weighted Model Counting (WMC), which we briefly review next. Consider a propositional logical formula $\phi$ over variables $\boldsymbol{v}$. Let $p$ be a probability vector over $\boldsymbol{v}$, with $p_i$ denoting the probability of the $i^{th}$ variable, $v_i$, being true. The WMC of formula $\phi$ under $p$ is then defined as:

$$\text{WMC}(\phi, \mathrm{p}) = \sum_{\omega \models \phi} \quad \prod_{i \in \omega} p_i \prod_{i \notin \omega} 1 - p_i. \tag{1}$$

In essence, the WMC is the sum of the probability of all worlds $\omega$ that are models of $\phi$. A widely-used approach for solving the WMC problem is *knowledge compilation* (KC) (Darwiche and Marquis, 2002; Chavira and Darwiche, 2008). The formula $\phi$ is first compiled into a tractable representation, which is used at inference time to compute a large number of queries - in our case, instances of the WMC problem - in polynomial time. KC techniques push most of the computational effort to the "off-line" compilation phase, resulting in computationally cheap "on-line" query answering, a concept termed *amortized inference*.

The representations obtained via KC are computational graphs, in which the literals, i.e., logical variables and their negations, are found only on leaves of the graph. The nodes are only logical AND and OR operations, and the root represents the query. To perform WMC, the boolean circuit is replaced by an arithmetic one, by replacing the AND nodes of the graph with multiplication, the OR nodes with addition, and the negation of literals with subtraction $(1 - x)$. Appendix A presents the application of KC on the running example.

## 2.3. Verification of Neural Networks

**NN Robustness.** Verifying the robustness of NN classifiers amounts to proving that the network's correct predictions remain unchanged if the input is perturbed within a given range $\epsilon$ (Wong et al., 2018). NN verification methods reason over infinitely-many inputs to derive formal certificates for the robustness condition. For a given network $f$, this is formalized as follows: for all inputs $\boldsymbol{x}$, such that $f(\boldsymbol{x})$ is a correct prediction, and for all $\boldsymbol{x}'$, such that $\|\boldsymbol{x} - \boldsymbol{x}'\| \leq \epsilon$, it holds that $f(\boldsymbol{x}) = f(\boldsymbol{x}')$. Checking if the robustness condition holds can be achieved by reasoning over the relations between the un-normalized predictions (logits) at the NN's output layer. In particular, if for any $\boldsymbol{x}'$ in an $\epsilon$-ball of $\boldsymbol{x}$, it holds that $y_{true} - y_i > 0$, for all $y_i \neq y_{true}$, then the network is robust for $\epsilon$ (Gowal et al., 2018). Here $y_{true}$ is the logit corresponding to the correct class and $y_i$ are the logits corresponding to all other labels. This condition can be checked by computing the minimum differences of the predictions for all points in the $\epsilon$-ball. If that minimum is positive, the robustness condition is satisfied. However, finding that minimum is NP-hard (Katz et al., 2017).

**Solver-Based Verification.** Early verification approaches include Mixed Integer Linear Programming (MILP) (Lomuscio and Maganti, 2017; Tjeng et al., 2019; Henriksen and Lomuscio, 2020) and Satisfiability Modulo Theories (SMT) (Ehlers, 2017b; Katz et al., 2017). MILP approaches encode the verification problem as an optimization task over linear constraints, which can be solved by off-the-shelf MILP-solvers. SMT-based verifiers translate the NN operations and the verification query into an SMT formula and use SMT solvers to check for satisfiability. Although these methods provide exact verification results, they do not scale to large, deep networks, due to their high computational complexity.

**Relaxation-Based Verification.** As the verification problem is NP-hard (Katz et al., 2017), incomplete techniques that do not reason over an exact formulation of the verification problem, but rather an over-approximating relaxation, are used for efficiency. A salient method that is commonly used is Interval Bound Propagation (IBP), a technique which uses interval arithmetic (Sunaga, 1958) to propagate the input bounds through all the layers of a NN (Gowal et al., 2018). As a non-exact approach to verification, it is not theoretically guaranteed to solve a problem. However, the approach is sound, in that if the lower bound is shown to be positive the network is robust. Therefore, once the bounds of the output layer are obtained, an instance is safe if the lower bound of the logit corresponding to the correct class is greater than the upper bounds of the rest of the logits, since this ensures a correct prediction, even in the worst case.

## 3. Probabilistic Neuro-Symbolic Robustness Verification

### 3.1. Problem Statement

We now formally define the aim of relaxation-based techniques in the context of NeSy reasoning systems. Given a NeSy system, as defined in Section 2.1, our aim is to compute:

$$\left[ \min_{\boldsymbol{x}'} p(y_i|\boldsymbol{x}'), \ \max_{\boldsymbol{x}'} p(y_i|\boldsymbol{x}') \right] \quad \forall \boldsymbol{x}' \ s.t. \ ||\boldsymbol{x}' - \boldsymbol{x}|| \leq \epsilon \tag{2}$$

for all $y_i$ in $\boldsymbol{y}$. That is, we wish to calculate the minimum and maximum value of each probabilistic output of the NeSy system, under input perturbations of size $\epsilon$. As described in Section 2.3, it is then possible to use these bounds to assess the robustness of an instance.

Consider the NeSy system of Figure 1. The neural part comprises two NNs: (1) an object detector predicting whether a red traffic light or a car is in front of the autonomous vehicle (AV), and (2) an action selector, which outputs whether to accelerate or brake the AV. The symbolic part is the conjunction of two constraints, as described in Appendix A. Given an input image $\boldsymbol{x}$, the system computes $y$, the probability that the constraints are satisfied. An instance is robust if $\min_{\boldsymbol{x'}} p(y|\boldsymbol{x'}) > T$, with threshold $T \in [0,1]$. This denotes that for all inputs in an $\epsilon$-ball of $\boldsymbol{x}$ the probability of the constraints being satisfied is always greater than $T$. Throughout the rest of the paper we consider the case where $T = 0.5$.

### 3.2. Exact Solution Complexity

Let us now assume that via techniques described in Section 2.3 we have obtained bounds in the form of a probability range for each output of the NN. Solving Equation 2 involves propagating these bounds through the symbolic component, in order to obtain maxima/minima on the reasoning output $\boldsymbol{y}$. We now analyze the complexity of performing this computation exactly. To do so, we introduce a decision problem, E-WMC ("exists" WMC). Contrary to the functional problem, which seeks the maximum/minimum WMC for a given set of variable bounds, E-WMC instead asks "is there a probability assignment within the given variable bounds, such that the WMC is at least T, for some given threshold T"? Formally:

**Definition 1 (E-WMC)** *Given a Boolean formula $\phi(\boldsymbol{v})$ over variables $\boldsymbol{v} = (v_1, \ldots, v_k)$, probability intervals $I_i = [l_i, u_i] \subseteq [0,1]$ for each variable $v_i$, and a threshold $T \in [0,1]$, is there a probability vector $w \in I_1 \times \ldots \times I_k$ such that $\mathrm{WMC}(\phi, w) \geq T$?*

Importantly, the computation involved in solving Equation 2 is at least as difficult as E-WMC, since (1) obtaining the maximum WMC value via Equation 2 allows to directly answer threshold queries of E-WMC, and further (2) E-WMC only involves the maximization subproblem of Equation 2.

**Proposition 2 (Complexity of E-WMC)** *E-WMC is $\mathrm{NP}^{\mathrm{PP}}$-complete.*
**Proof** *For membership, we notice that the circuits comprising the symbolic component represent multi-linear polynomials of the input variables (Choi et al., 2020). As all input variables of the polynomial are defined in a closed interval, the extrema lie on the vertices of the domain (Laneve et al., 2010), i.e. each variable is assigned either its lower or upper bound, not something in between. This yields a combinatorial search space of $2^n$ possible assignments. Checking if $\mathrm{WMC}(\phi, w) \geq T$ for each assignment can be solved via a #P-query (Chavira and Darwiche, 2008). Thus, we need to search in a combinatorial space in which each guess is linear in the size of the input (the NP-part) with a call to a #P-oracle at each step, which establishes that E-WMC is in $\mathrm{NP}^{\#\mathrm{P}}$. From (Monniaux, 2022) $\mathrm{NP}^{\#\mathrm{P}} = \mathrm{NP}^{\mathrm{PP}}$.*

*To show hardness (see Appendix B for a full proof), we reduce E-MAJSAT (Littman et al., 1998), the SAT-oriented complete problem for $\mathrm{NP}^{\mathrm{PP}}$, to E-WMC. Specifically, we show that given any formula $\phi$ we can find a threshold $T$ and a set of probability intervals $I$ for the variables of $\phi$, such that E-MAJSAT$(\phi) \Longleftrightarrow$ E-WMC$(\phi, I, T)$.*  ∎

### 3.3. Relaxation-Based Approach

The hardness of exact bound computation through the compiled symbolic component motivates the use of relaxation-based techniques. We now show how these can be extended to the NeSy setting in order to provide a scalable solution to Equation 2.

In the NeSy systems of Section 2.1, the outputs of the NN are fed as input to the arithmetic circuit. Due to this compositionality and the algebraic nature of the circuits, a NeSy system can be seen as an end-to-end differentiable computational graph. Such a graph can be constructed as a single module within a machine learning library, such as Pytorch, and subsequently exported as an Open Neural Network Exchange (ONNX) graph (developers, 2021). Appendix C depicts the ONNX representation of the NeSy system of the running example, shown as an end-to-end algebraic graph.

ONNX is the standard input format for NN verifiers (Brix et al., 2024), including both solver-based verification tools, such as Marabou (Katz et al., 2019), and relaxation-based ones, such as auto_LiRPA (Xu et al., 2020) and VeriNet (Henriksen and Lomuscio, 2020). Thus, by exporting a NeSy system to this format, it is possible to utilize state-of-the-art tools to perform verification in an almost "out-of-the-box" fashion. While our proposed framework is, in principle, compatible with all the aforementioned tools, we focus on relaxation-based verifiers, in order to showcase scalable probabilistic NeSy verification. Such verifiers allow us to perturb the input and compute bounds directly on the output of the NeSy system, that is, without computing intermediate bounds on the NN outputs.

## 4. Experimental Evaluation

In this section we empirically evaluate the effectiveness and applicability of our approach. We assess the scalability of the proposed method via a synthetic task based on MNIST addition, a standard benchmark from the NeSy literature (Manhaeve et al., 2018). Further, we apply our approach to a real-world autonomous driving dataset and verify a safety driving property on top of two 6-layer convolutional NNs. In this case, the scalability of our technique allows us to handle high-dimensional input and larger networks, which are typical of real-world applications. All experiments are run on a machine with 128 AMD EPYC 7543 32-Core processors (3.7GHz) and 400GB of RAM.

### 4.1. Multi-Digit MNIST Addition

In this experiment we evaluate the scalability of our approach as the complexity of the symbolic component increases. Specifically, we explore how the approximate nature of our method enhances scalability, while also considering the corresponding trade-off in the quality of verification results. To this end, we compare the following approaches:

1. **End-to-End relaxation-based verification** $(\text{E2E-R})$
   An implementation of our method in auto_LiRPA , a state-of-the-art relaxation-based verification tool. The input to auto_LiRPA is the NeSy system under verification, which is translated internally into an ONNX graph. The verification method used is IBP, as implemented in auto_LiRPA .

2. **Hybrid verification** $(\text{R+SLV})$
   A hybrid approach consisting of relaxation-based verification for the neural part of

the NeSy system and solver-based bound propagation through the symbolic part. The former is implemented in auto_LiRPA using IBP. The latter is achieved by transforming the circuit into a polynomial (see Section 3.2), and solving a constrained optimization problem with the Gurobi solver. The purpose of comparing to this baseline is to assess the trade-off between scalability and quality of results, when using exact vs approximate bound propagation through the symbolic component.

3. **Solver-based verification** (Marabou)
Exact verification using Marabou, a state-of-the-art SMT-based verification tool, also used as a backend by most NeSy verification works in the literature (Xie et al., 2022; Daggitt et al., 2024). Marabou is unable to run on the full NeSy architecture, as the current implementation[2] does not support several operators, such as Softmax and tensor indexing. To obtain an indication of Marabou's performance, we use it to verify only the neural part of the NeSy system, a subtask of NeSy verification. Specifically, we verify the robustness of the CNN performing MNIST digit recognition.

**Dataset.** We use a synthetic task, where we can controllably increase the size of the symbolic component, while keeping the neural part constant. In particular, we create a variant of multi-digit MNIST addition (Manhaeve et al., 2018), where each instance consists of multiple MNIST digit images, and is labelled by the sum of all digits. We can then control the number of MNIST digits per sample, e.g. for 3-digit addition, an instance would be ( (▨, ▨, ▨), 13). We construct the verification dataset from the 10K samples of the MNIST test set, using each image only once. Thus, for a given #digits the verification set contains 10K/#digits test instances.

**Experimental setting.** The CNN[3] recognizes single MNIST digits. It is trained in a supervised fashion on the MNIST train dataset (60K images) and achieves a test accuracy of 98%. The symbolic part consists of the rules of multi-digit addition. It accepts the CNN predictions for the input images and computes a probability for each sum. As the number of summands increases, so does the size of the reasoning circuit, since there are more ways to construct a given sum (e.g. consider the ways in which 2 and 5 digits can sum to 17). We vary the number of digits and the size of the perturbation added to the input images. We consider five values for #digits: $\{2, 3, 4, 5, 6\}$ and three values for the perturbation size $\epsilon$: $\{10^{-2}, 10^{-3}, 10^{-4}\}$, resulting in 15 distinct experiments. For each experiment, i.e., combination of #digits and $\epsilon$ values, we use a timeout of 72 hours. E2E-R runs on a single thread, while the Gurobi solver in R+SLV dynamically allocates up to 1024 threads.

**Scalability.** Figure 2 presents a scalability comparison between the methods. The figure illustrates the time required to verify the robustness of the NeSy system for a single sample, averaged across the test dataset. All experiments terminate within the timeout limit, with the exception of two configurations for R+SLV. For $\langle \epsilon = 10^{-2}, \#\text{digits} = 5, 6 \rangle$, R+SLV was not able to verify any instance within the timeout (which is why the lines for $\epsilon = 10^{-2}$ stop at 4 digits). For $\langle \epsilon = 10^{-3}, \#\text{digits} = 6 \rangle$, R+SLV verifies less than 5% of the examples within the timeout. The reported values in Figure 2 are the average runtime for this subset.

---

2. https://github.com/NeuralNetworkVerification/Marabou
3. The CNN comprises 2 convolutional layers with max pooling, 2 linear layers, and a softmax activation.

As Figure 2 illustrates, E2E-R scales exponentially better than R+SLV – note that runtimes are in log-scale. This is due to the computational complexity of exact bound propagation through the probabilistic reasoning component, as shown in Section 3.2. In verifying the robustness of the CNN only, Marabou's runtime is 314 seconds per sample, averaged across 100 MNIST test images. It is thus several orders of magnitude slower than our approach, in performing a subtask of NeSy verification.

This indicative performance aligns with theoretical (Zhang et al., 2018) and empirical (Brix et al., 2024) evidence on the poor scalability of SMT-based approaches. Our results suggest that this trade-off between completeness and scalability is favourable in the NeSy setting, which may involve multiple NNs and complex reasoning.

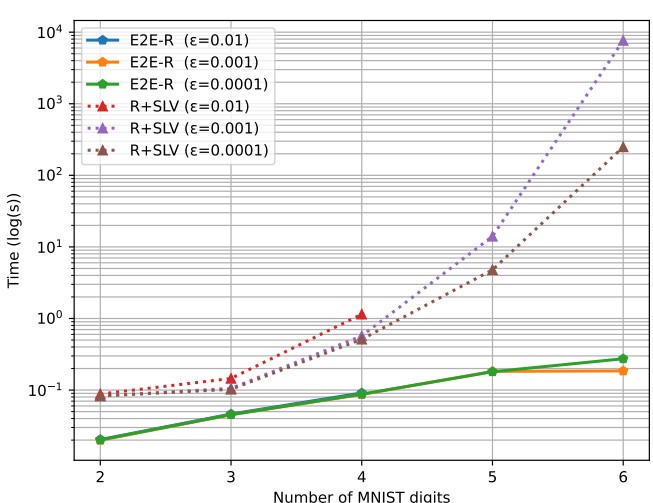

**Quality of verification results.** We next investigate how the complexity of the logic affects the quality of the verification results. In Table 1 we report: (a) the tightness of the output bounds, in the form of lower/upper bound intervals for the probability of the *correct* sum for each sample, averaged across the test set; (b) the robustness of the NeSy system, defined as the the number of robust samples divided by the total samples in the test set.

Figure 2: Comparison of verification runtime as the number of summand digits increases. We report the time required to verify the NeSy system on a single sample, averaged across the MNIST test dataset, and repeat the experiment for 3 values of the perturbation size.

Table 1: Comparison of performance as the number of summand digits increases, $\epsilon = 0.001$. We report one metric for bound tightness and one metric for robustness. Metrics for 6 digits are omitted since the full experiment exceeds the timeout.

| Verification Method | Metric | #MNIST digits | | | |
|---|---|---|---|---|---|
| | | 2 | 3 | 4 | 5 |
| R+SLV | Lower/Upper Bound | $0.871 - 0.981$ | $0.815 - 0.972$ | $0.764 - 0.962$ | $0.731 - 0.928$ |
| | Robustness (%) | 90.60 | 86.17 | 81.33 | 78.31 |
| E2E-R | Lower/Upper Bound | $0.871 - 0.982$ | $0.815 - 0.974$ | $0.763 - 0.965$ | $0.716 - 0.958$ |
| | Robustness (%) | 90.60 | 86.11 | 81.21 | 76.67 |

As expected, R+SLV outputs strictly tighter bounds than E2E-R for all configurations. We further observe that the quality of the bounds obtained by E2E-R degrades as the size of the reasoning circuits increases. This is also expected, since errors compound and accumulate over the larger end-to-end graph. However, the differences between R+SLV and E2E-R are minimal, especially in terms of robustness.

### 4.2. Autonomous Driving

In this experiment we apply our proposed approach to a real-world dataset from the autonomous driving domain. The purpose of the experiment is to assess the robustness of a neural autonomous driving system with respect to the safety and common-sense properties of Figure 1, i.e., to evaluate whether input perturbations cause the neural systems to violate the constraints that they previously satisfied.

**Dataset.** To that end, we use the ROad event Awareness Dataset with logical Requirements (ROAD-R) (Giunchiglia et al., 2023). It consists of 22 videos of dashcam footage from the point of view of an autonomous vehicle (AV), and is annotated at frame-level with bounding boxes. Each bounding box represents an *agent* (e.g. a pedestrian) performing an *action* (e.g. moving towards the AV) at a specific *location* (e.g. right pavement).

**Experimental Setting.** We focus on a subset of the dataset that is relevant to the symbolic constraints of Figure 1. Consequently, we select a subset of frames which adhere to these constraints. Specifically, either the AV is moving forward, there is no red traffic light in the frame, and no car stopped in front of the AV, or the AV is stopped, and there is either a red traffic light or a car stopped in front. We sample the videos every 2 seconds to obtain a dataset of 3143 examples. Each example contains a $3 \times 240 \times 320$ image, and four binary labels: red_light, car_in_front, stop, move_forward. The neural part of the system comprises two 6-layer CNNs[4], responsible for object detection and action selection respectively. The two networks are trained in a supervised fashion using an 80/20 train/test split over the selected frames. The object detection and action selection networks achieve accuracies of 97.2% and 96.3% on the respective test sets. We perturb the test input images using five values of perturbation size $\epsilon$: $\{10^{-5},\ 5 \cdot 10^{-5},\ 10^{-4},\ 5 \cdot 10^{-4},\ 10^{-3}\}$.

Table 2: Autonomous driving experiment results, indicating robustness and verification runtime for five values of the $\epsilon$-perturbation.

| Metric | Epsilon | | | | |
|---|---|---|---|---|---|
| | 1e-5 | 5e-5 | 1e-4 | 5e-4 | 1e-3 |
| Robustness (%) | 96.82% | 92.68% | 82.64% | 6.21% | 0.00% |
| Runtime per Sample (s) | 0.091 | 0.092 | 0.091 | 0.092 | 0.092 |

Table 2 presents the results. We report robustness, the fraction of robust instances over the total number of instances in the test set, and verification runtime for E2E-R. Since this task consists of a small arithmetic circuit and a significantly larger neural component, it is the latter that predominantly affects both the computational overhead and the accumulated errors of bound propagation. Therefore, E2E-R and R+SLV, which differ only in the symbolic component, provide nearly identical results that are omitted. As expected, robust accuracy falls as the perturbation size increases. Regarding the verification runtime, this experiment reinforces our results from Section 4.1, by demonstrating that the runtime of our approach remains largely unaffected by changes in the value of the perturbation size $\epsilon$.

---

4. The CNNs have 4 convolutional layers with max pooling and 2 linear ones. The object detection network has a sigmoid activation at the output, while the action selection network has a softmax.

## 5. Related Work

Verifying properties on top of hybrid systems combining neural and symbolic components remains largely under-explored, with few related verification approaches existing in the literature. Akintunde et al. (2020) address the problem of verifying properties associated with the temporal dynamics of multi-agent systems. The agents of the system combine a neural perception module with a symbolic one, encoding action selection mechanisms via traditional control logic. The verification queries are specified in alternating-time temporal logic, and the corresponding verification problem is cast as a mixed-integer linear programming (MILP) instance, delegated to a custom, Gurobi-based verification tool. Xie et al. (2022) seek to verify symbolic properties on top of a neural system. The authors introduce a property specification language based on Hoare logic, which allows for trained NNs, along with the property under verification, to be compiled into a satisfiability modulo theories (SMT) problem. This is then delegated to Marabou, a state-of-the-art SMT-based verification tool. Daggitt et al. (2024) follow a similar approach, in order to verify NeSy programs, i.e., programs containing both neural networks and symbolic code. The authors introduce a property specification language, which allows for NN training and the specification of verification queries. A custom tool then compiles the NNs, the program, and the verification query into an SMT problem, which is again delegated to Marabou.

These aforementioned approaches differ substantially from our proposed method, since they cannot verify probabilistic logical reasoning systems. This is because they utilize formalisms of limited expressive power (logics of limited expressive power in (Akintunde et al., 2020; Xie et al., 2022) and a functional language in (Daggitt et al., 2024)), and which lack a formal probabilistic semantics. In contrast, our method is agnostic to the choice of knowledge representation framework, and can verify the robustness of any NeSy system which represents symbolic knowledge as an algebraic computational graph. For example, the proposed method supports the full expressive power of logic programming under probabilistic semantics by compiling logic programs into arithmetic circuits via KC, as is typically the case with state-of-the-art probabilistic NeSy systems (Manhaeve et al., 2018; Yang et al., 2023). Furthermore, all existing approaches are based on solver-based verification techniques, which translate the verification query into an SMT (Xie et al., 2022; Daggitt et al., 2024) or a MILP problem (Akintunde et al., 2020). While such approaches are sound and complete, they suffer from serious scalability issues, which often renders them impractical.

## 6. Conclusion

We presented a scalable technique for verifying the robustness of probabilistic neuro-symbolic reasoning systems. Our method combines relaxation-based techniques from the NN verification domain with knowledge compilation, in order to assess the effects of input perturbations on the probabilistic logical output of the system. We motivated our approach via a theoretical analysis, and demonstrated its efficacy via experimental evaluation on synthetic and real-world data. Future work includes extending our method to more sophisticated NN verification techniques, such as (Reverse) Symbolic Interval Propagation (Gehr et al., 2018; Wang et al., 2021), and certified training techniques (Müller et al., 2023; Palma et al., 2024) towards increasing the estimated robustness of the system.

## Acknowledgments

This work is supported by the project EVENFLOW – Robust Learning and Reasoning for Complex Event Forecasting, which has received funding from the European Union's Horizon research and innovation programme under grant agreement No 101070430. Alessio Lomuscio is supported by a Royal Academy of Engineering Chair in Emerging Technologies.

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

## Appendix A. Knowledge Compilation Example

Consider the constraints $\phi$ from the autonomous driving example of Figure 1:

$$\text{red light} \vee \text{car in front} \implies \text{brake}$$
$$\text{accelerate} \iff \neg\text{brake}$$

These dictate that (1) if there is a red light or a car in front of the AV, then the AV should brake, and (2) that accelerating and braking should be mutually exclusive and exhaustive, i.e., only one should take place at any given time. Figure 3(a) presents the compiled form of these constraints as a boolean circuit, namely a Sentential Decision Diagram (SDD) (Darwiche, 2011).

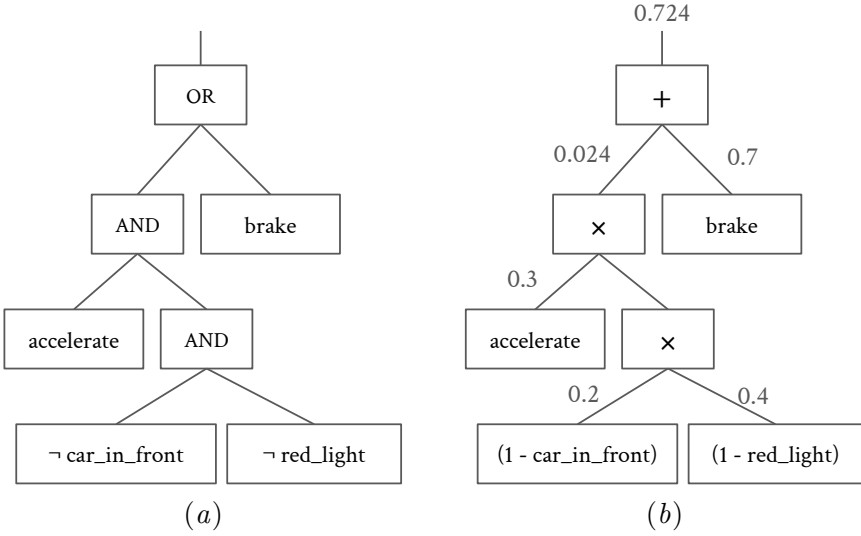

Figure 3: (a) A Sentential Decision Diagram (SDD) as an example of a computational graph obtained via knowledge compilation and (b) the corresponding arithmetic circuit (AC) derived from the SDD during inference by replacing the AND/OR nodes with multiplication/addition. The SDD has been minimized for conciseness.

By replacing the AND nodes of the graph with multiplication, the OR nodes with addition, and the negation of literals with subtraction $(1 - x)$ (the NAT semiring (Maene and De Raedt, 2025)), one obtains an arithmetic circuit (AC), shown in Figure 3(b). The resulting structure can compute the WMC of $\phi$ simply by plugging in the literal probabilities at the leaves and traversing the circuit bottom-up. Indeed, one can check that assuming the probabilities:

$$p(\text{accelerate}) = 0.3, \quad p(\text{red light}) \quad = 0.6$$
$$p(\text{brake}) = 0.7, \quad p(\text{car in front}) = 0.8$$

this computation correctly calculates the probability of $\phi$ by summing the probability of its 5 different models.

## Appendix B. Hardness of E-WMC

**Proposition 3**  E-WMC *is* $\text{NP}^{\text{PP}}$-*hard.*

**Proof**  *We give a reduction from* E-MAJSAT *(Littman et al., 1998), the SAT-oriented complete problem for* $\text{NP}^{\text{PP}}$, *to* E-WMC.

**Definition 4 (E-MAJSAT)**  *Given a Boolean formula* $\phi(\boldsymbol{x}, \boldsymbol{y})$ *over variables* $\boldsymbol{x} \cup \boldsymbol{y}$, *where* $\boldsymbol{x} = (x_1, \ldots, x_n)$ *and* $\boldsymbol{y} = (y_1, \ldots, y_m)$, *is there an assignment* $x \in \{0, 1\}^n$ *such that the majority of assignments to* $\boldsymbol{y} \in \{0, 1\}^m$ *satisfy* $\phi(x, \boldsymbol{y})$? *We denote the restriction of* $\phi$ *to an assignment* $x$ *as* $\phi|_x$. *Further, we denote as* $\#\phi|_x$ *the number of satisfying assignment of* $\boldsymbol{y}$, *such that* $\phi|_x(y) = \text{true}$. *Formally,* E-MAJSAT *is given by:*

$$\exists\, x \in \{0, 1\}^n : \#\phi|_x \geq \frac{2^m}{2}$$

*We show that there exists a threshold* $T^*$ *and a set of probability intervals* $I^*$ *such that* E-MAJSAT$(\phi) \Longleftrightarrow$ E-WMC$(\phi, I^*, T^*)$. *Specifically, consider the construction:*

$$T^* = \frac{1}{2}\,, \qquad I_i^* = \begin{cases} [0,\ 1] & v_i \in \boldsymbol{x} \\ [1/2,\ 1/2] & v_i \in \boldsymbol{y} \end{cases}$$

*Thus, we allow variables in* $\boldsymbol{x}$ *to assume any weight, while we set the weights of all variables in* $\boldsymbol{y}$ *to* $1/2$. *Consider now a weight assignment to* $\boldsymbol{x}$ *(for example* $p(x_1) = 0.7, p(x_2) = 0.6$*), which induces a distribution* $p(\boldsymbol{x})$. *We then have:*

$$\text{WMC}(\phi, p) = \sum_{x\ \in\ \{0,1\}^n} p(x) \cdot \frac{1}{2^m} \cdot \#\phi|_x \tag{3}$$

*That is, rather than calculating the WMC by summing the probability over all models of* $\phi$, *we instead sum over sets of worlds, corresponding to each partial truth assignment* $\{0, 1\}^n$ *to the variables* $\boldsymbol{x}$. *The probability of each such set is the product of (1) the probability of* $x$ *under this assignment (as dictated by* $p(\boldsymbol{x})$*), (2) the probability of* $y$ *(which is always* $m$ *variables of weight* $1/2$*), and (3) the number of worlds in this subset that are models of* $\phi$.

($\Longrightarrow$)  *If* E-MAJSAT$(\phi)$ *is true then* E-WMC$(\phi, I^*, T^*)$ *is also true. Let the solution of* E-MAJSAT *be* $x^*$. *Consider now a weight assignment to* $\boldsymbol{x}$ *equal to* $p(\boldsymbol{x} = x) = \mathbf{1}[x = x^*]$. *That is, assign to each variable* $x_i \in \boldsymbol{x}$ *a probability equal to the truth value of* $x_i$ *in* $x^*$. *Further, set the weight of all variables* $y_i \in \boldsymbol{y}$ *to* $1/2$. *In this case, the sum of Equation 3 will reduce to a single term, the one corresponding to* $x = x^*$. *By construction,* $p(x) = 1$ *for this term, and* $p(x) = 0$ *for all other terms. It then follows immediately that:*

$$\text{WMC}(\phi, p) = \frac{1}{2^m} \cdot \#\phi|_{x^*} \geq \frac{1}{2^m} \cdot \frac{2^m}{2} = \frac{1}{2}$$

*since* E-MAJSAT$(\phi)$ *is true by assumption. Hence, there exists a weight vector* $w^*$ *within* $I^*$ *such that* $\text{WMC}(\phi, w^*) \geq 1/2$, *and so* E-WMC$(\phi, I^*, T^*)$ *is true.*

($\Longleftarrow$)   *If* E-WMC*($\phi$, $I^*$, $T^*$) is true then* E-MAJSAT*($\phi$) is also true. We prove this by contradiction. Assume that* E-MAJSAT*($\phi$) is false, that is, $\forall x : \#\phi|_x < 2^m/2$. Then:*

$$\text{WMC}(\phi, p) = \sum_x p(x) \cdot \frac{1}{2^m} \cdot \#\phi|_x \;<\; \sum_x p(x) \cdot \frac{1}{2} \;<\; \frac{1}{2} \sum_x p(x) \;<\; \frac{1}{2}.$$

*By assumption,* E-WMC*($\phi$, $I^*$, $T^*$) is true and so* $\text{WMC}(\phi, p) \geq 1/2$*, and we have reached a contradiction. Thus: $\exists x : \#\phi|_x \geq 2^m/2$ making* E-MAJSAT*($\phi$) true.* ∎

## Appendix C. NeSy ONNX Representation

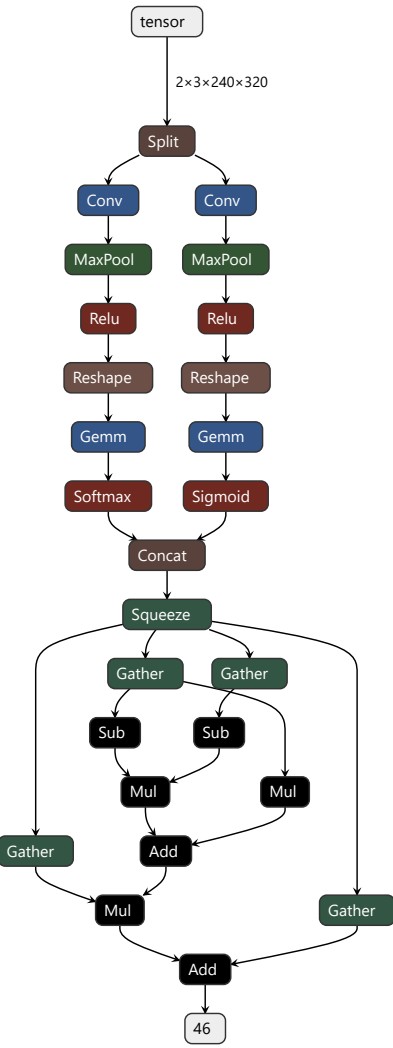

Figure 4: Unified ONNX representation of the NeSy system of the running example. The input image is processed by the two NNs (left branch is action selection, right branch is object detection) and then through the arithmetic circuit. The NNs are stripped down to one convolutional layer (Conv + MaxPool + ReLU) and one dense layer (Reshape + Gemm + Softmax/Sigmoid) for conciseness. The operators in the circuit, besides Add, Sub, and Mul, are created by Python operations, such as tensor indexing and concatenation.

