# OpenReview forum: "A Scalable Approach to Probabilistic Neuro-Symbolic Robustness Verification"
_nesyconf.org/NeSy/2025/Conference_Phase_2 — NeSy 2025 - Phase 2 Oral_

### Official Review · Reviewer_3efV · 2025-06-28
**Scalable and Principled Verification for Probabilistic Neuro-Symbolic Systems**

**Rating:** 7
**Confidence:** 2

**Review:**

This paper presents a scalable verification approach for probabilistic neuro-symbolic (NeSy) systems. The authors analyze the complexity of verifying robustness properties in such systems and prove that the decision version of the verification task is NP^PP-complete. In response to this hardness result, they propose an approximate, relaxation-based method that compiles the entire NeSy system—comprising both neural and symbolic components—into a computational graph suitable for standard NN verification tools. Extensive experiments on synthetic (multi-digit MNIST addition) and real-world (autonomous driving) tasks demonstrate the scalability and practicality of the approach.

The paper is well-motivated and makes a solid contribution by addressing a previously under-explored but important verification problem in probabilistic NeSy systems. The complexity-theoretic analysis adds rigor, and the proposed integration with ONNX-based verifiers is both technically sound and practically useful.

That said, some aspects could be strengthened:
- The discussion of circuit size and structure in knowledge compilation remains shallow—while noted as future work, preliminary insights or measurements would have been valuable.
- While the symbolic component remains precise during standard, unperturbed inference, the broader implications of using approximate verification methods, e.g., the trustworthiness of explanations derived from verification outputs, remain further explored. A brief discussion or a case study (e.g., via failure diagnosis) could clarify when such approximations are practically acceptable.

**Anonymity:**

Disclose identity

---

### Official Review · Reviewer_whDF · 2025-07-04
**Fixed completeness proof, NP^PP membership remains broken. Practically still of interest.**

**Rating:** 7
**Confidence:** 4

**Review:**

The manuscript has significantly improved over the previous version. And in particular the problem with the NP^PP hardness has been addressed.

However the NP^PP completeness proof still exhibits the same blasé engagement with computational complexity theory as the previous draft. The NP^PP membership part of the proof contains false statements and is missing some necessary details.

1) NP is the class of problems solvable in polynomial time on a non-deterministic turing machine. That means you can make *polynomial size* guesses and then verify them in polynomial time. If you need to make larger guesses, that would take superpolynomial time (as it is implemented as superpolynomially many non-deterministic steps in the TM). While the authors have correctly noticed that there are only 2^n possible guesses, it is not clear from the current argument that these guesses are in fact all of polynomiall bounded size. That is, the lower/upper bounds that are guessed must all representable in polynomial size of the input.

2) PP is not simply the decision version of #P. Under normal assumptions #P is more powerful, for example you can solve parityP problems with with a way to solve #P, but PP and parityP are incomparable. However the mistake here is easily fixed since the paper of Monniaux I pointed the authors to in the last review shows that NP^#P = NP^PP. So it's ok to simply use the #P oracle without the erronous claim about the relationship to PP.

I trust that the authors are able to address this in their final draft and my score is to be understood this assumption.

Other than this my positive notes on the practical aspects of the work from my last review stand and I find the presentation and organisation of this revision is a clear improvement over the previous version.

**Anonymity:**

Remain anonymous

---

### Official Review · Reviewer_DDJu · 2025-07-08
**The paper has improved**

**Rating:** 9
**Confidence:** 4

**Review:**

The authors have responded to many of the points raised in phase 1. This is an interesting and well written paper on a topic that has until now been understudied in a NeSy context. I'm glad that the suggestion of "verification" has been removed and been replaced by the more accurate "robustness verification", clarifying the actual contribution.

The paper is overall well written, with a novel problem (at least novel in the NeSy context), a good motivation, a nice (and simple) approach (where "simple" is a feature, not a bug), and two experiments that support the point of the paper.

**Anonymity:**

Disclose identity